# Deciphering the antigen specificities of antibodies by clustering their complementarity determining region sequences

Dianita S. Saputri,[1,2] Hendra S. Ismanto,[1] Dendi K. Nugraha,[3] Zichang Xu,[4] Yasuhiko Horiguchi,[2,3,5] Shuhei Sakakibara,[4,6] Daron M. Standley[1,2,4,5]

**ABSTRACT** Recent advances in adaptive immune receptor repertoire sequencing have provided abundant B cell receptor (BCR) sequences under various conditions, including vaccination and disease. However, determining target antigen and epitope specificity of the corresponding antibodies is a major challenge due to their exceptional sequence diversity. Here, we introduce a novel method to cluster antibodies sharing antigenic targets based on their complementarity determining region (CDR) sequences. Using the proposed method, we demonstrate that SARS-CoV-2 spike protein receptor-binding domain (RBD) binders and non-RBD binders from publicly available BCR data were classified correctly, with a cluster purity of 95%. These clusters were then leveraged for annotating unlabeled COVID-19 patient BCR data, enabling the discovery of novel anti-RBD antibodies. We further validated the method by clustering BCR repertoires obtained from single-cell immune profiling of diphtheria-tetanus-pertussis (DTP)-vaccinated donors. Antibody expression and antigen-binding assays demonstrated that the clusters exhibited 96% antigen purity, surpassing the apparent 82% purity achieved by assigning antigens to the same B cells using fluorescently labeled DTP antigen probes. Moreover, antibodies within certain clusters were found to possess neutralizing activity, suggesting that CDR clusters contain epitope-level information. Together, this study offers a simple approach for antigen- and epitope-specific BCR discovery that is reproducible, inexpensive, and applicable to a wide range of antigen targets.

**IMPORTANCE** Determining antigen and epitope specificity is an essential step in the discovery of therapeutic antibodies as well as in the analysis adaptive immune responses to disease or vaccination. Despite extensive efforts, deciphering antigen specificity solely from BCR amino acid sequence remains a challenging task, requiring a combination of experimental and computational approaches. Here, we describe and experimentally validate a simple and straightforward approach for grouping antibodies that share antigen and epitope specificities based on their CDR sequence similarity. This approach allows us to identify the specificities of a large number of antibodies whose antigen targets are unknown, using a small fraction of antibodies with well-annotated binding specificities.

**KEYWORDS** antibody, antigen-specific, B cell receptor, clustering, diphtheria, pertussis, tetanus, repertoire, SARS-CoV-2, single-cell immune profile

Address correspondence to Daron M. Standley, standley@biken.osaka-u.ac.jp.

We wish to disclose that we are in the process of applying for a patent describing the antibody clustering method.

See the funding table on p. 15.

Adaptive immune responses, which include activation of B and T cells, provide specific and long-lasting protection against various pathogens (1). This remarkable ability is due to the extraordinary diversity of B cell receptors (BCRs) and T cell receptors (TCRs), which are generated combinatorially from V, D, and J gene fragments (1–3). Due to the nature of V(D)J recombination, adaptive immune responses are highly variable among individuals. However, at present, there is no consensus about the best way to

assess adaptive immune responses across donors using immune receptor repertoires. Meanwhile, adaptive immune receptor repertoire (AIRR) sequencing has revealed a broad range of immune responses to specific pathogenic antigens by sampling the BCRs or TCRs from circulating blood or disease-related tissues. Classifying AIRRs by their target antigens and epitopes could be a promising approach to compare adaptive immune response of different donors.

As the volume of immune receptor sequence data continues to increase, there have been numerous attempts to infer antigen specificities computationally. In the case of TCRs, sequence-based clustering has been extensively employed in order to define peptide-major histocompatibility (pMHC) specificity groups (4–7). In contrast, deciphering antigen specificity of BCRs (denoted as antibodies in their soluble form) has been more challenging. Unlike TCRs, which recognize pMHCs in a structurally restricted manner, antibodies recognize antigen surface patches, resulting in a greater range of possible conformational epitopes. On the antibody side, residues that interact with epitopes are known as paratopes, and these primarily occur in or near three complementarity-determining regions (CDRs 1–3) (8, 9). Current BCR clustering methods predominantly rely on the concept of a "clonotype," which is a group of receptors that share the same V and J genes as well as possess highly similar CDR3 amino acid sequences (10–15). While clonotypes are expected to target the same antigen and epitope, mature antibodies from distinct genetic origins can also converge to target the same antigen and epitope (16, 17). Moreover, during the affinity maturation process, BCRs undergo extensive somatic hypermutation in the variable regions, including CDRs 1–3 (1, 18). For these reasons, clonotyping alone may be less sensitive in functional clustering of BCRs into antigen-specific groups.

Previously, we found that BCRs targeting the same or similar conformational epitopes could be clustered by their CDR sequences (19). Another study has similarly reported that the use of predicted paratope sequences resulted in more sensitive antibody clustering than clonotyping (20). CDR sequences have also been found to facilitate the elucidation of antibody-antigen binding modes by docking (21, 22). These observations suggest that clusters of CDR sequences may be the preferred way of defining BCR specificity groups.

Here, we introduce a CDR sequence-based clustering approach that is simple, intuitive, and easy to implement using established sequence analysis tools (23). We validate the specificity of this approach using antibodies with known binding specificities to SARS-CoV-2 spike protein from CovAbDab (24). We then leverage these antibodies to assign the specificities of previously published BCR sequence data from COVID-19 patients. We also applied the clustering method to new BCR sequence data obtained from diphtheria-tetanus-pertussis (DTP)-vaccinated donors and discover novel neutralizing antibodies clusters. Our study demonstrates that the CDR cluster is sensitive and reliable method for assigning antigen specificity to antibody repertoire data.

## RESULTS

### Representing BCRs by paratope pseudo sequences

Since paratopes are primarily composed of residues within CDRs (21), we can describe BCRs by using a pseudo sequence composed of the three concatenated CDRs. Typical pseudo sequences are long enough to provide sufficient alignment coverage at rigorous sequence identity thresholds, thereby providing pairwise scores that are statistically significant, which is not the case for CDR3 sequence alignments, in general. Here, the InterClone code base was used to prepare pseudo sequences and to call the MMseqs2 cluster method (23) in order to group the pseudo sequences efficiently (Fig. 1), as described in Materials and Methods.

### Validation of coronavirus antigen domain-specific antibody clusters

To validate the effectiveness of the CDR clustering approach, we utilized the publicly available BCR database, CovAbDab, which contains a vast collection of BCRs with known

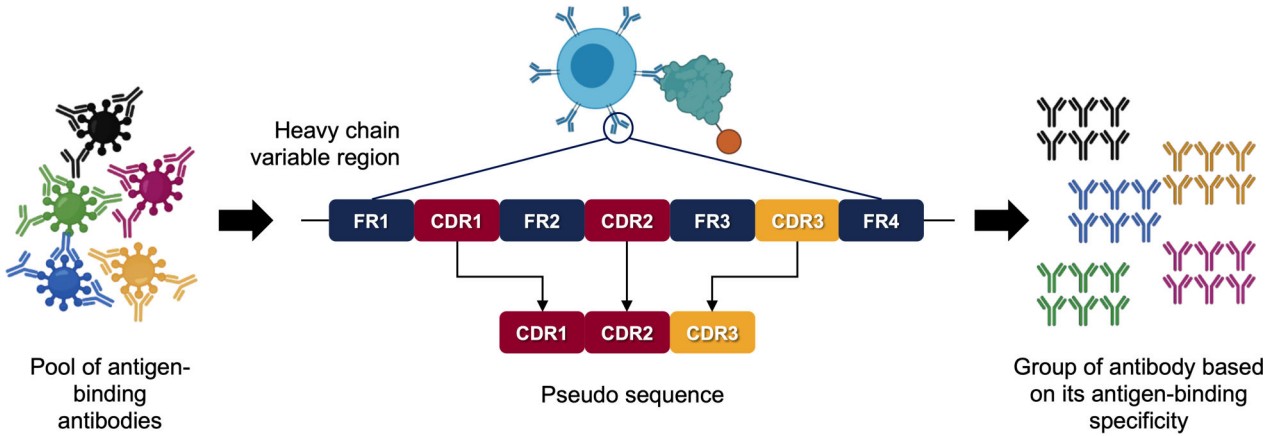

**FIG 1** CDR clustering approach (modified from reference 25). CDRH1, CDRH2, and CDRH3 amino acid sequence were retrieved and sequence identity cutoff was determined. Antibodies that have coverage and sequence identity above threshold will be clustered.

specificity to coronavirus antigens (24). From this data set, we excluded non-human antibodies and focused on 6,412 antibodies recognizing the SARS-CoV-2 spike protein receptor-binding domain (RBD), as well as 3,702 "non-RBD" antibodies targeting other regions of the spike protein (Fig. 2A). Following pseudo sequence preparation, we performed clustering using several sequence identity and coverage thresholds. To understand how well the sequences within a cluster match a particular annotation (in this case, either RBD or non-RBD), we calculated the "cluster purity," which is defined as the fraction of members annotated with the most frequent assignment (RBD, non-RBD) within the cluster (26). As shown in Fig. 2B; Table S1, the mean cluster purity was sensitive to both coverage and sequence identity. Coverage threshold was set to a value of 90% in order to reduce the possibility of alignment across CDR boundaries. Sequence identity threshold was then adjusted to achieve a cluster purity of 95% or higher. By using 90% coverage and 80% sequence identity threshold, we obtained a cluster purity of 95.3% among 1,074 non-singleton clusters, 25 of which are shown in Fig. 2C. Alignment of the CDRH sequences among the top three largest clusters indicates that CDRH3 sequences were the least conserved, as expected, but that there was variation even in CDRs 1–2 (Fig. 2D). We note that the mean pairwise CDRH3 sequence identities within clusters are approximately 10% lower than the typical lower bound for clonotyping (80%) suggesting that the CDR clusters can group antibodies from different clonal groups sharing paratope antigen specificity.

## Identification of novel anti-RBD antibodies by co-clustering unlabeled antibody sequences from COVID-19 patients with CovAbDab data

Next, we assessed the efficacy of the CDR clustering method to identify novel anti-RBD antibodies within unlabeled BCR sequence data (Fig. 3A). To this end, we clustered the validated SARS-CoV-2-specific antibodies from CovAbDab with unlabeled single-cell BCR sequence data obtained from a previous study of COVID-19 patients (28). Using the same sequence identity and coverage thresholds established above, we found that 1,863 out of 52,799 unlabeled BCR sequences from COVID-19 patients clustered with validated anti-SARS-CoV-2 antibodies from CovAbDab, resulting in 602 clusters. The top 25 largest clusters that include unlabeled BCRs are shown in Fig. 3B. The co-clustering of labeled and unlabeled data indicates that CDR clustering can detect "public" antibody responses after SARS-CoV-2 infection. We then expressed five unlabeled antibodies belonging to four clusters including those from RBD Clusters 1 and 3 and from non-RBD Clusters 4 and 6. We evaluated the binding specificity of the unlabeled antibodies to SARS-CoV-2 full-length spike protein and RBD-expressing Expi293F cells. All antibodies tested bound to the spike protein. Antibodies from Clusters 1 and 3, but not Cluster 4 or 6, exhibited

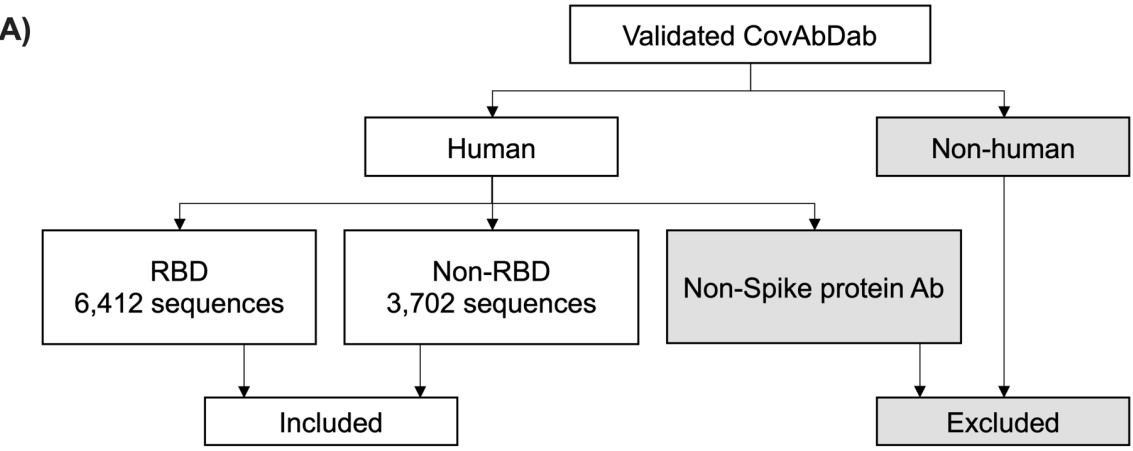

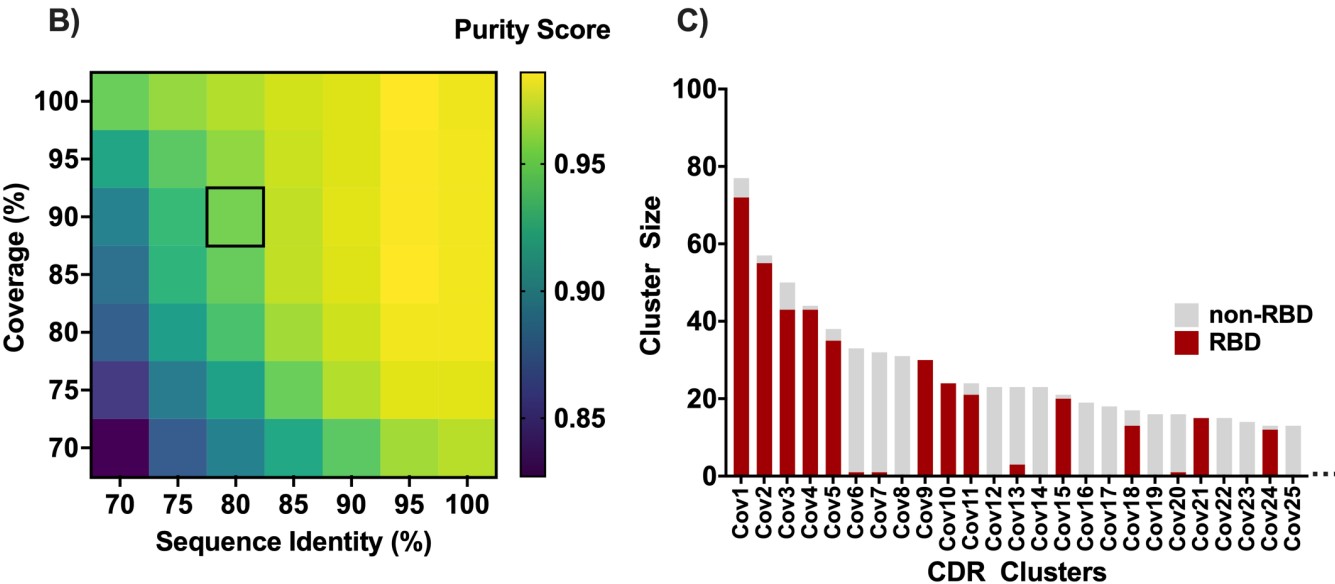

FIG 2 CDR cluster antigen specificity in validated BCR database. (A) Collecting BCR data from CovAbDab. (B) Cluster purity score comparison among various coverage and sequence identity thresholds. Square indicates the chosen cluster purity (95.3%), obtained by 80% and 90% of sequence identity and coverage thresholds, respectively. The detail cluster purity scores are also shown in Table S1. (C) The 25 largest clusters of CovAbDab data, colored by the antigen specificity, for which the cluster purity score was 95%. (D) Multiple sequence alignments of CDRHs among top three biggest clusters made by WebLogo (27). The cluster size (*n*) and sequence identities for CDRs1–3 are also shown.

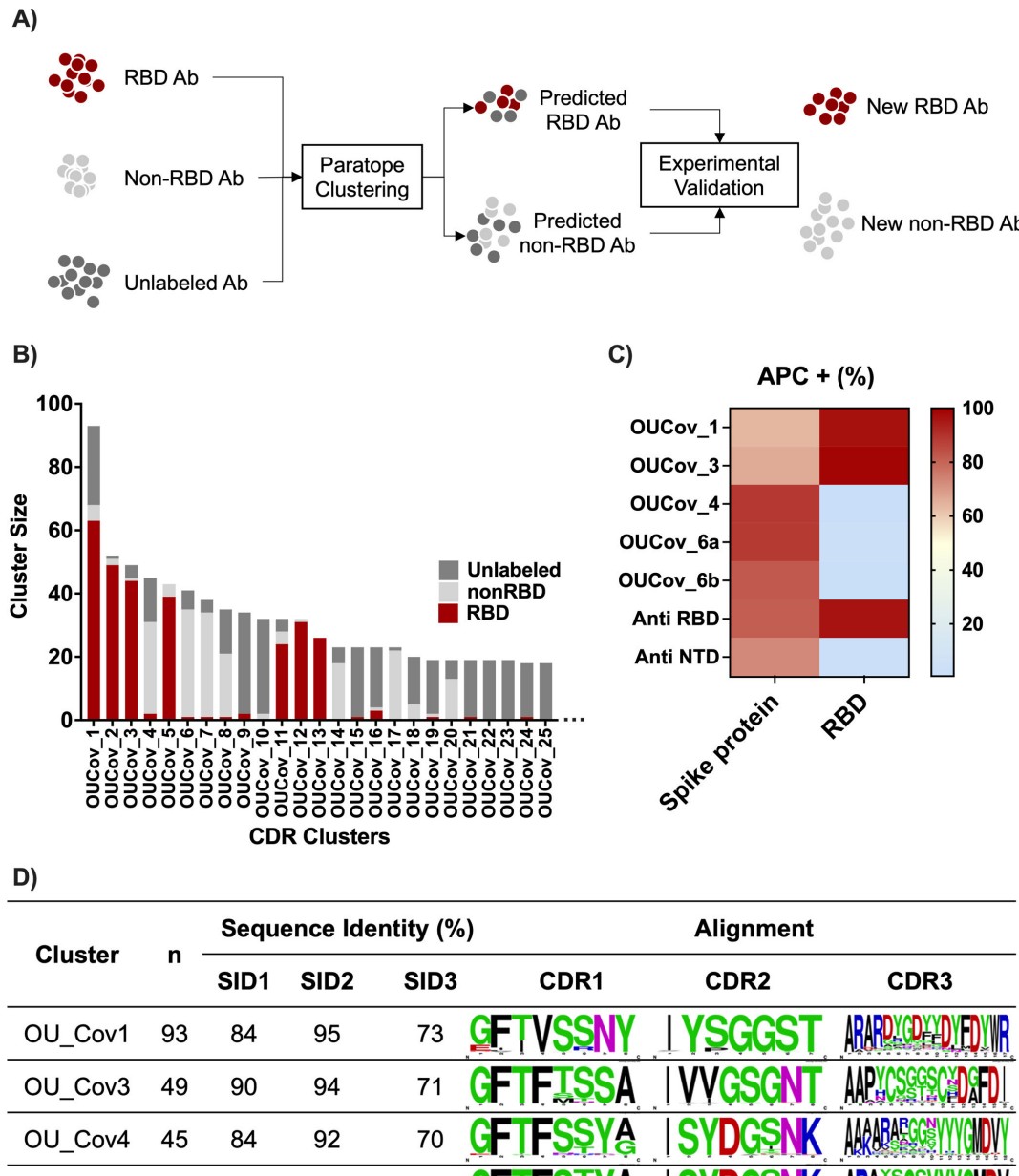

**FIG 3** CDR cluster of validated and non-validated (unlabeled) BCR data. (A) Study design of CDR clustering to annotate unlabeled antibodies. (B) Clustering of validated CovAbDab data with unlabeled antibody data from COVID-19 patients in Osaka University (OU) identified potential RBD and non-RBD binding antibodies (dark gray presents in the same cluster with red or light gray). (C) Experimental binding assay of unlabeled antibody representatives from Cluster OUCov 1, 3, 4, and 6. Values represent percent population (mean ± SD) of spike protein or RBD-expressing cells that positively bound to antibody, detected by APC-conjugated anti-human IgG. (D) CDRHs sequence alignment of OUCov1, 3, 4, and 6, made by WebLogo (27). The cluster size (*n*) and sequence identities for CDRs1–3 are also shown.

binding to the RBD, confirming the predicted assignment of their binary (RBD, non-RBD) labels (Fig. 3C). Alignment of the CDRHs among the validated clusters indicated highly but not perfectly conserved CDRH1 and CDRH2 sequences, and much greater variability in CDRH3 (Fig. 3D). This finding demonstrated that CDR clustering is effective in assigning target antigens of unlabeled human BCR repertoires using a limited set of labeled antibody data.

## CDR clustering of BCRs derived from vaccinated donors identifies poly-clonal clusters

The unprecedented breadth and depth of CovAbDab make COVID-19 the exception rather than the rule among infectious diseases in terms of antigen-labeled BCR sequence data. We next aimed to assess the utility of the CDR clustering method in the annotation of antigens associated with other diseases. We choose the DTP vaccine as an immune perturbation model because of its proven efficacy to provide robust and broad protection to three highly lethal pathogens: diphtheria, tetanus, and pertussis. To this end, we collected BCR repertoire data from four healthy donors vaccinated with the trivalent DTP vaccine. All donors were adult volunteers with a childhood history of DTP vaccination (Table 1) and received a single DTP booster shot. We collected peripheral blood mononuclear cells (PBMCs) 7 days after booster administration (Fig. S1A). We sorted $CD19^+CD27^+$ antigen-experienced B cells and further isolated four distinct subgroups of B cells by employing fluorescently labeled DTP toxins/toxoids as antigen probes. Three B cell subgroups consisted of those that bound to diphtheria toxin (DT), tetanus toxin (TT), or pertussis toxin (PT), respectively. The fourth subgroup (NB) consisted of those that did not bind to any of these antigens (Fig. S1B). The four groups of cells were then subjected to 10× Genomics single-cell immune profiling. Following quality control procedures, we obtained a total of 13,847 paired antibody sequences, comprising both heavy and light chains, for further analyses (Fig. 4A).

We then conducted both clonotype and CDR clustering of the paired BCR sequences, resulting in a total of 11,636 clonotype and 10,802 CDR clusters. CDR clustering yielded more non-singleton clusters compared to clonotyping as shown in Fig. 4B; Table S2 ($P$ < 0.0001). Of the 4,243 BCR sequences belonging to 1,198 non-singleton clusters, we found 69 clusters with members assigned to multiple V genes, 46 clusters with multiple J genes, and 1 cluster with members assigned to multiple V and J genes (Fig. 4A). In total, 116 (9.68%) clusters were poly-clonal, that is, composed of multiple VDJ clonotypes. These results highlight the difference between clonotyping and CDR-based clustering, with the latter revealing the convergence from distinct clonotypes to functionally similar paratopes in antibody response after DTP vaccination. Nevertheless, clonotype- and CDR-based clusters showed a comparable proportion of public clusters (28.6% and 32.72%, respectively) (Fig. 4C; Table S3, $P$ <0.0001), suggesting that the use of CDRs did not result in over-clustering.

## Experimental validation of new bacterial-toxin specific antibodies

To validate antigen specificity, we analyzed the 25 largest DTP clusters. We provisionally assigned antigens to the BCRs based on the antigen probe (DT, TT, PT, or NB) they bound to during the cell sorting process. Unexpectedly, we observed the presence of clusters whose members bound to different antigen probes (Fig. 5A). The mean antigen cluster purity using these provisional assignments was only 82%, much lower than the 95% observed in the COVID-19 data analysis. Possible explanations for the disagreement between the CDR clustering and probe-based antibody assignment include (i) non-specific binding of fluorophores, streptavidin, or antigen purification tags causing a lenient gating strategy and high false-positive rate; (ii) sticky or polyreactive BCRs; (iii) and decreased cluster purity when applied to raw BCR data without any validated antibodies.

To experimentally address these three scenarios, we selected 1–4 non-identical antibody sequences with different provisional antigen assignments from each of the 25

**TABLE 1** Subject demographics

| ID | Sex | Age (years old) | Ethnicity |
|---|---|---|---|
| Donor 1 | Female | 30 | Asian |
| Donor 2 | Female | 29 | Asian |
| Donor 3 | Female | 27 | Asian |
| Donor 4 | Male | 34 | Asian |

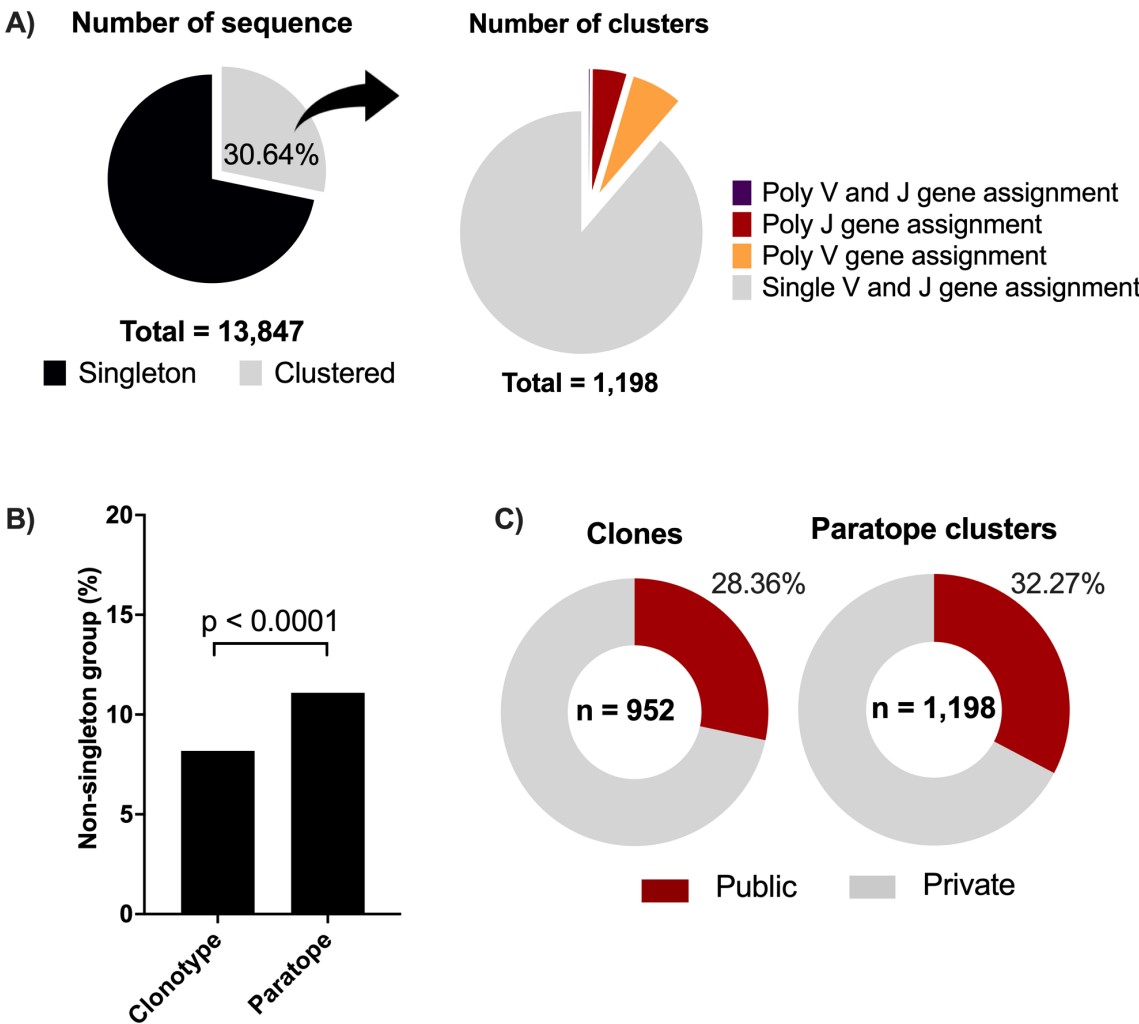

**FIG 4** Frequency of polyclonal cluster. (A) Proportion of singleton and clustered antibodies. A total of 4,243 antibody sequences were clustered into 1,198 distinct groups. Among 1,198 clusters, 116 (9.68%) were polyclonal clusters where 69, 46, and 1 cluster have more than one V, J, and V/J gene assignment, respectively. (B) Comparison of non-singleton group appearance in CDR cluster and clonotype. Statistical difference was analyzed by Chi Square test (see Table S2). (C) Observation of public clones and public paratope clusters, defined as total number of public clone or cluster divided by total number of clone or cluster, respectively. Statistical difference was analyzed by chi square test (see Table S3).

largest clusters (Fig. 5B). A total of 52 naturally paired BCR sequences were expressed as human IgG1 monoclonal antibodies. Indirect ELISA was performed to assess the binding of these antibodies to each DTP toxoid (DT, TT, PT), as well as to the SARS-CoV-2 spike protein as an unrelated antigen. Post-vaccination serum was employed as a positive control in these experiments. Out of the 52 recombinant antibodies derived from the top 25 largest clusters, 10 antibodies did not exhibit reactivity toward any of the three antigens, while 16, 2, and 24 antibodies exhibited specific binding to DT, TT, or PT, respectively (Fig. 5C). Although a few antibodies within Clusters 9 and 13 tested negatives, none of the clusters contained members that tested positive to more than one antigen. Thus, the corrected cluster purity was, in fact, found to be 96% within the limitations of the sampling performed (Fig. 5C and D), which is consistent with the COVID-19 assessment above. In total, we identified 7, 1, and 8 clusters with binding specificity to DT, TT, and PT, respectively. These results validated the antigen-specificity of the CDR clusters and further identified novel DTP-binding antibodies, potentially possessing neutralizing activities. By mapping back to all members of the 25 largest

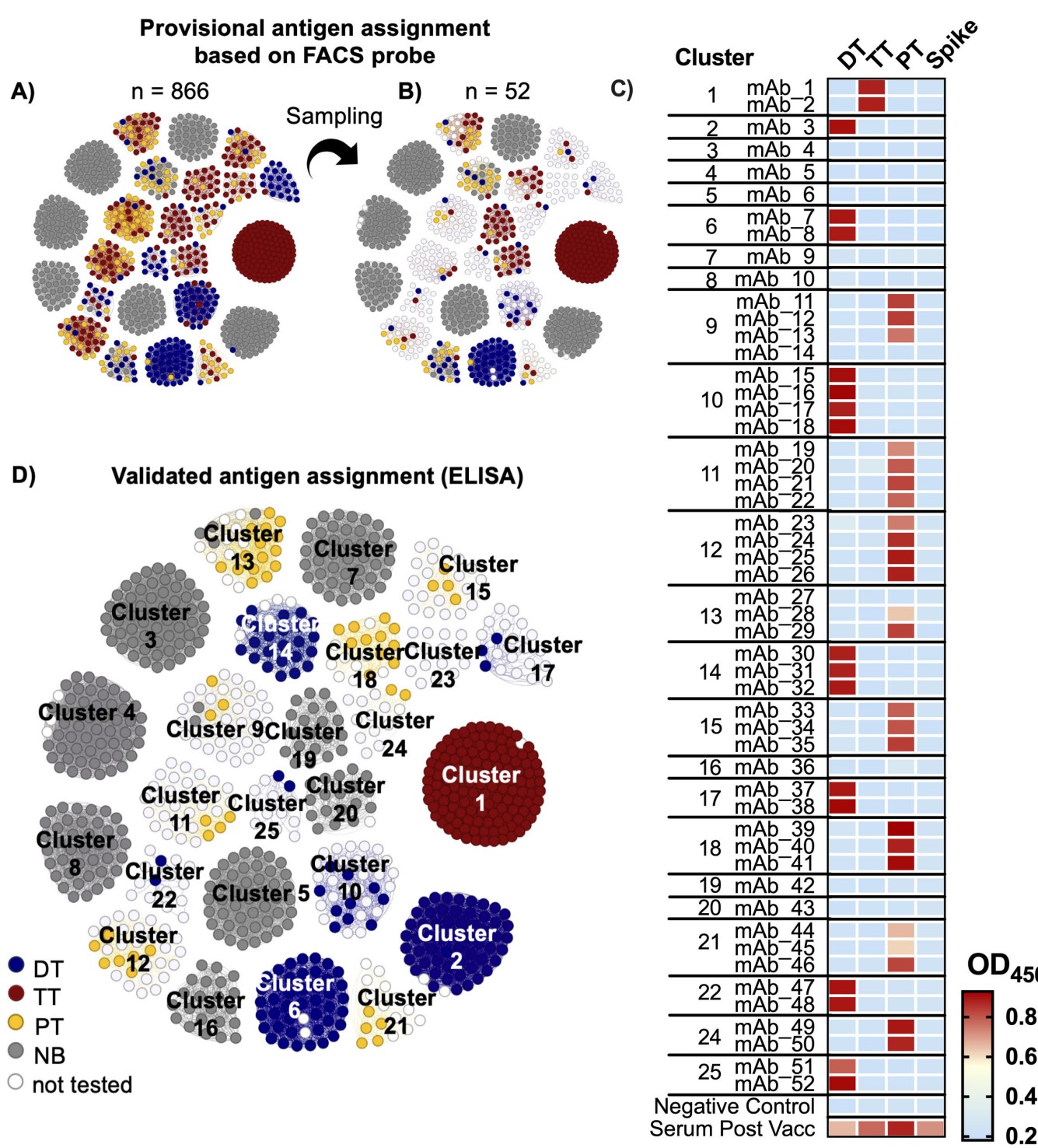

**FIG 5** CDR clustering of DTP-vaccinated repertoire. (A) Visualization of top-25 biggest clusters with provisional antigen assignment based on cell-sorting probes. One small circle represents one BCR sequence. Cluster purity score was 82%. Some clusters have members that bound to different antigen probes during the cell-sorting process. (B) Representative sequences were selectively chosen for validation, with a particular emphasis on those exhibiting provisional antigen assignments from diverse antigen probes. In total, we tested 52 non-identical sequences from Clusters 1–25. (C) Heatmap of ELISA-binding assay to DT, TT, PT, and SARS-CoV-2 spike protein. Value represents $OD_{450}$ from duplicated wells. (D) Corrected network graph after validation. Cluster purity score was 96%. Network figure in A, B, and D was generated using Gephi Software.

clusters, the assessment of 52 recombinant antibodies enabled the specificities of over 800 previously uncharacterized antibodies to be elucidated.

Additionally, by examining the DTP antigen specificities on a per-donor basis (Fig S2), it was readily apparent that all of the donors except Donor 3 produced antibodies in response to all three antigens. These results were qualitatively consistent with serum-level assays, which indicated that all donors except Donor 3 showed an increase in antibody responses to all three antigens (Fig S3). Indeed, among the expressed antibodies from Clusters 1–25, Donor 3 only contributed to clusters that did not bind to DTP antigen (Cluster 3, 7, 20), indicating that Donor 3 is a weak responder to the vaccine. These results suggested that CDR clustering not only identified numerous antigen-specific clones, including ones representing public antibody responses to a multivalent vaccine, but also provided an efficient roadmap for assigning the antigen specificities of these BCRs experimentally as well as comparing the adaptive immune responses across multiple donors.

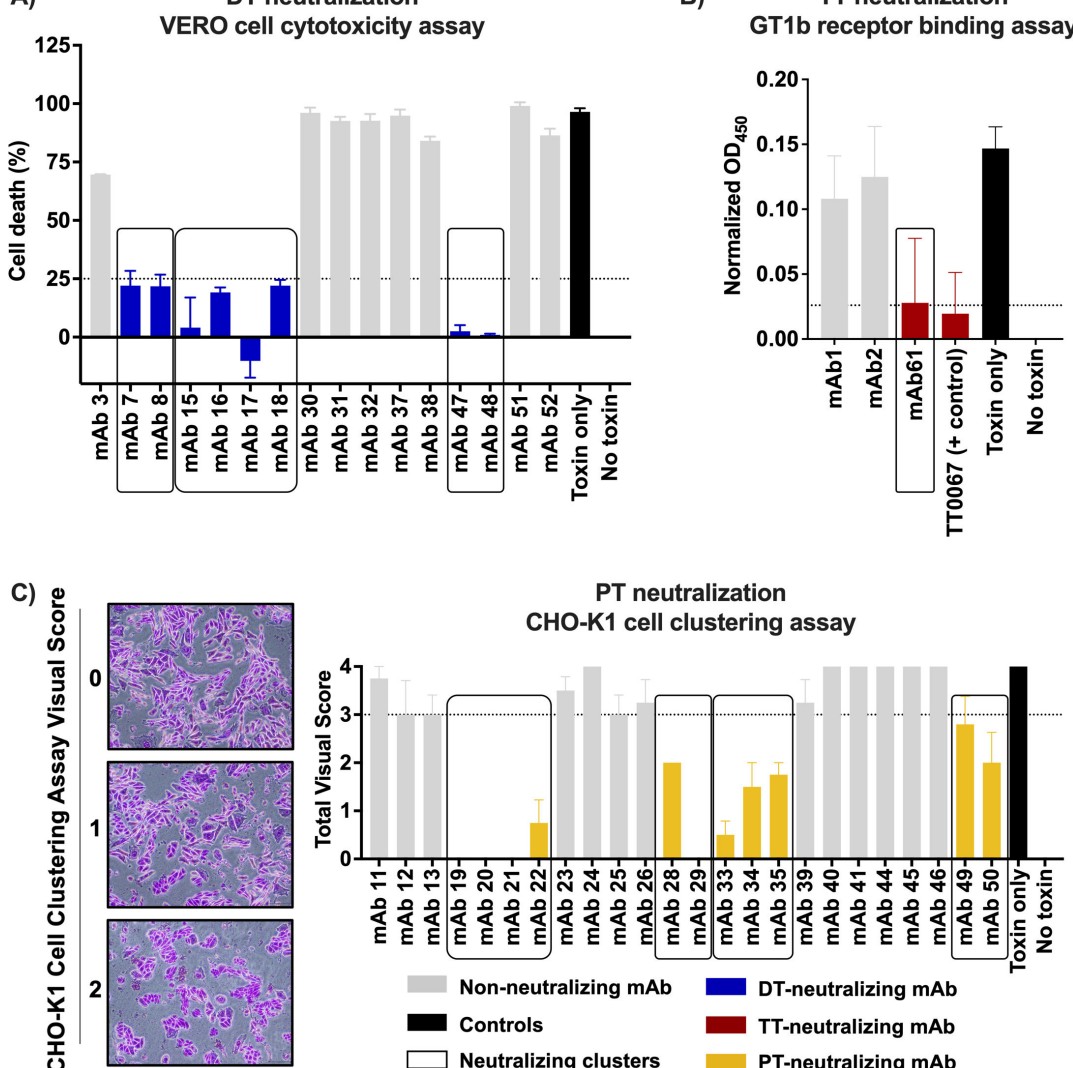

**FIG 6** Neutralization assay of DTP-binding antibodies. (A) VERO cell cytotoxicity assay. To test the antibody inhibition effect on VERO cell death, $5 \times 10^3$ cells/well were incubated with DT in the presence or absence of antibodies. After 24 h incubation, cell death was quantified as detailed in Materials and Methods. Values represent the mean ± SD from duplicated wells. (B) TT-GT1b receptor binding inhibition assay by ELISA. Values represent mean ± SD of normalized $OD_{450}$ in duplicated wells. (C) CHO-K1 cell clustering assay. To score each well: 0, very minor cells clustered (<30%); 1, equivocal; 2, majority of cells clustered. Visual score was obtained from total score of duplicated wells. Five observers gave score independently and the mean ± SD was determined and plotted. Total visual score less than 3 was considered a positive neutralization effect of antibodies.

## *In vitro* assays identified novel DTP-neutralizing antibodies within the same cluster

To further assess the functional specificity of the DTP-specific clusters, we conducted three distinct *in vitro* neutralization assays. We performed a cytotoxicity assay to evaluate the inhibitory effect of DT-binding antibodies on VERO cell death induced by DT (29, 30). Notably, 3 out of 7 DT-binding clusters (comprising 8 out of 16 DT-binding antibodies) exhibited a reduction in cell death (Fig. 6A). Subsequently, we employed an ELISA assay to investigate the ability of TT-binding antibodies to inhibit TT binding to the GT1b receptor (31, 32). No significant impact on TT binding to the GT1b receptor was observed for the antibodies in Cluster 1. We then expanded the search for TT-binding and neutralizing antibodies to smaller DTP clusters and found a TT binding antibody, mAb61, in Cluster 40 that inhibited TT binding to the receptor at a comparable level to the positive control, TT0067 (33) (Fig. 6B). Finally, to examine neutralization by PT-binding antibodies, we utilized the CHO-K1 cell line, which undergoes morphological changes upon exposure to PT (34) (Fig. 6C). Remarkably, four out of eight PT-binding clusters (encompassing 11 out of 24 PT-binding antibodies) neutralized the toxin's effect on CHO-K1 cells (Fig. 6C). Overall, we observed functional consistency (neutralizing or non-neutralizing) within a given cluster. This indicates that the clusters contain epitope-level information in addition to antigen specificity.

## DISCUSSION

Assessing the antibody and epitope specificity of antibodies elicited by a particular immune perturbation is of paramount importance for the development of therapeutic antibodies and the study of immune responses to disease or vaccination (35). However, this is a non-trivial task due to the diverse nature of antibody responses across donors (1–3). Here, we presented a novel approach to decipher the antigen specificities of B cell responses to vaccination or infection by clustering antibodies by their CDR sequences. CDR clustering using a panel of annotated antibodies against SARS-CoV-2 spike protein resulted in high antigen cluster purity in terms of two categories of spike binders (RBD vs non-RBD). We further showed that the clusters could be used to assign domain-level specificities of new unlabeled BCR data from COVID-19 patients, which highlights the ability of the clustering method to leverage annotated BCR repertoire data.

Compared to clonotyping, where highly similar CDR3 amino acid sequences and identical V-J assignments are used to group antibodies (10–15), CDR clustering at a pseudo sequence identity threshold of 80% resulted in clusters with mean pairwise CDR3 sequence identities less than 80%, suggesting that the CDR clusters represent a degree of convergence of different antibody clonal groups that target the same antigen and epitope (16, 17). This finding is also supported by DTP data analysis where CDR clustering defined public antibodies and non-singleton clusters at a higher rate than clonotype clustering, resulting in more sequences that belong to different clones to converge into shared clusters. We also note that, by using paratope rather than clonotype as representation of BCRs, we can include the effects of somatic hypermutation changes in the clusters. Therefore, features based on paratopes may allow prediction of antibody-antigen affinity. For this reason, CDR clustering can be used in tandem with clonotyping in order to improve clustering across clones and across donors.

By using CDR clustering along with minimal antigen screening assays, we elucidated the antigen specificities of dominant B cell responses to DTP vaccination in multiple donors. Interestingly, the distribution of CDR cluster frequencies was unique for each donor in spite of the overall similarity in their serum-level responses. Moreover, we were able to identify a weak responder to the vaccine, highlighting the heterogeneity of human antibody responses to the same immune perturbation. It will be interesting to examine the cluster frequencies of larger number of donors in order to obtain robust signatures of vaccination and disease. This results also highlight the potential of CDR cluster to compare and evaluate the adaptive immune responses to a given vaccine among different donors.

It is a common practice to evaluate B cell responses to antigens using fluorescence-activated cell sorting (FACS). However, the DTP analysis suggested that provisional assignment of antigen specificities using FACS can itself be a source of error. Although antigen sorting could differentiate between positive and negative B cells, antigen specificities were less reliable, probably due to the non-specific binding of fluorophores, streptavidin, or antigen purification tags (35, 36). Double-fluorescent antigen-tagging techniques (37) or conjugation with sequencing-readable barcodes, as demonstrated in LIBRA-seq (38), have the potential to increase the confidence in antigen-specificity. However, even with such techniques, the possibility of obtaining false-positive sorted cells is not insignificant. Along with these experimental techniques, CDR clustering can help mitigate cell sorting errors by grouping receptors with highly similar paratopes resulting in antigen-specific clusters.

Clustering also assisted in the identification of neutralizing antibodies. Paratope clustering may potentially be used to identify antibody clusters that target antigens associated with different viral or bacterial strains. In the future, it will be interesting to examine whether such antibody clusters can serve as biomarkers for broad neutralization upon vaccination. By performing neutralizing assays for a small subset of cluster representatives, we efficiently identified and characterized potentially therapeutic antibodies against DTP infection from human PBMCs. Not all DTP-binding antibodies exhibited neutralizing properties, presumably due to the presence of non-neutralizing epitopes within clusters of antigen-positive BCRs. However, neutralizing activities were consistent within particular clusters, suggesting that the CDR clusters themselves were not merely antigen specific but also contained epitope-level information. While further investigations are needed to determine the exact epitopes targeted by these antibodies, the current approach of grouping by CDRs followed by verifying representatives for antigen reactivity and function is an efficient strategy for investigating antibody responses under any conditions. This approach will have applications to the study of adaptive immunity states associated with health and disease as well as for screening functional antibodies from a large number of candidates.

## MATERIALS AND METHODS

### Collecting COVID-19 BCR data from public database

Labeled human antibodies that known to bind spike protein (RBD or non-RBD) were downloaded from CovAbDab Database (24) with inclusion and exclusion criteria explained in Fig. 2A. Unlabeled COVID-19 antibody data were obtained from previous study in Osaka University (28). Raw sequencing data of that study are available at the Japanese Genotype-phenotype Archive (JGA) with accession codes JGAS00059/JGAD000722.

### Obtaining human samples of DTP vaccinated repertoires

This study was approved by Osaka University Institutional Review Board. Informed consent was obtained from the participants after providing detailed information about the research objectives and potential implications. Four healthy adults, aged between 25 and 35 years, were recruited and received a 0.5-mL intramuscular boost of DTP vaccine (TRIBIK, BIKEN). Blood samples were collected on 7 days after the vaccine shot. PBMCs were isolated from the blood samples using Histopaque-1077 (Sigma Aldrich) and 50-mL-Leucosep Tube (Greiner-BioOne) following the manufacturer's instruction and cryopreserved in CellBanker1 (Zenogen Pharma) until further use. The blood serum samples were also collected before and 7 days after shot and stored in 2× dilution with PBS.

## Serum IgG level measurement by ELISA

Capture antigens (DT, TT, PT [BIKEN]) were coated onto half-area 96-wells microplates and incubated overnight at 4°C. After two washes, the plate was blocked with 1% BSA in PBS. Following two washes, plates were then incubated with blood serum diluted 1,000 times in PBS. After three times washing, HRP-conjugated anti-human IgG (Jackson ImmunoResearch) were added and incubated for 40 min. After two washes, substrate in TMBZ (Bakelite Sumitomo) was added to each well for 10 min, and the reaction was stopped using $H_2SO_4$ solution (Bakelite Sumitomo). The experiments were performed in duplicates, and the $OD_{450}$ was measured using GloMax Explorer Multimode Microplate Reader (Promega).

## Generation of fluorescent-labeled antigen probes

The DT CRM-197 gene, a non-toxic mutant of DT, was amplified from a plasmid (a gift from Professor Eisuke Mekada) and cloned into the pCold II expression vector with 6× His-Tag at the N-terminal. The gene encoding the TT-binding domain was amplified from *Clostridium tetani* genomic DNA (a gift from Professor Tetsuya Iida) and cloned into the same expression system. These plasmids were then transformed into *Escherichia coli* BL21 and induced with IPTG at a final concentration of 0.5 mM for expression. The toxins were purified using AKTA Pure HisTrap $Ni^{2+}$ affinity chromatography (Cytiva). The His-Tag is preserved and used to label the toxins by using APC-conjugated anti-His antibody (BioLegend). Purified PT (BIKEN) was stained using Alexa Fluor 647 Protein Labeling Kit (Invitrogen) according to manufacturer's protocol.

## Cell sorting to get toxin-binding and DTP non-binding B cells

Cryopreserved PBMCs were rapidly thawed in a 37°C water bath for 1 min and then washed with 10 mL cold PBS. After centrifugation, the supernatant was removed, and the PBMCs were resuspended in PBS supplemented with 2% FBS (Cytiva). PBMCs were incubated with Fc-blocker (BioLegend) followed by staining with an antibody cocktail containing anti CD19-AF488 (BioLegend), anti CD3-BV421 (BioLegend), anti CD27-PE (BioLegend), and anti CD38-APC.Cy7 (BioLegend) for 30 min together with fluorescence-labeled antigen probes. Additionally, four different Total-Seq C anti-human hashtag antibodies (BioLegend) were used to label the donor. After three washes, cells from different donors were combined in FACS buffer (2% FBS in PBS) and sorted using BD Aria II Cell Sorter, with the gating strategy described in Fig. S1B. Antigen-experienced B cells ($CD19^+CD27^+$) were sorted, and double negative controls were utilized to gate the toxin- and non-binding cells: unlabeled toxin and the non-conjugated anti-His APC (BioLegend) for DT and TT, or non-conjugated Alexa Fluor 647 (Invitrogen) for PT.

## Sequence data collection of DTP repertoire

The library construction and sequencing of DTP-vaccinated repertoire were carried out at the Osaka University Genome Informatic Research Center Facility. The VDJ library was prepared using the 10× Chromium System (10× Genomics). In brief, we have four samples annotated as DT, TT, PT-sorted B cells, and NB (not bound to DTP), each of which contains mixture of samples from four different donors labeled by Total Seq C anti-human Hashtag antibody. Raw data processing and initial quality control were performed using Cell Ranger 6. The AIRR file outputs from Cell Ranger were used as an input file for the CDR-clustering.

## CDR clustering

The CDR pseudo sequence definition (Fig. 1A) employed in this study is similar to that described previously (19, 25). In brief, the BCR sequences in AIRR formatted file were subjected to annotation and quality control. The resulting sequences were then processed to extract the CDRH1, CDRH2, and CDRH3 amino acid sequences, which

were subsequently concatenated as a single paratope pseudo sequence. These pseudo sequences were grouped using predetermined thresholds of coverage and sequence identity using the MMseqs2 software (23). Here, the coverage threshold was set to 0.9 using coverage mode 0, which requires the alignment to include at least 90% of both aligned sequences. We used a high threshold in order to preserve CDR boundaries. The sequence identity cutoff was 0.8. After clustering the sequence based on these thresholds, the clusters were sorted based on size and visualized using GraphPad Prism 9 for the COVID-19 data and Gephi for DTP data.

## Antibody expression

The antibody variable regions, including the heavy and light chain pairing genes, were synthesized by Integrated DNA Technologies. Recombinant antibodies were generated using established protocols as previously described (25). Antibodies were purified using protein A spin column (CosmoBio), and their concentrations were determined by ELISA using hIgG standard curve.

## COVID-19 antigen-binding assay by flow cytometry

To assess antibody binding to the SARS-CoV-2 antigen, we followed the methodology described in a previous study (25). The binding of antibodies to the SARS-CoV-2 antigen-expressing cells was analyzed using flow cytometry (Attune NxT, Thermo).

## DTP-antigen-binding assay by ELISA

Capture antigens (DT, TT, PT [BIKEN]), including SARS-CoV-2 spike protein (a gift from Professor Masato Okada) as a control, were coated onto half-area 96-wells plate and incubated overnight at 4°C. The subsequent steps followed the same procedure as the assessment of serum IgG response with one difference in the sample application step. In this case, 25 µL/well of a 1-µg/mL solution of recombinant antibody was applied and incubated for 2 h at room temperature. Post-vaccination serum was used as the positive control.

## VERO cell-DT antibody neutralization assay

The VERO cell line (a gift from Professor Eisuke Mekada) was cultured in DMEM (Wako) supplemented with MEM non-essential amino acid (Wako), 10% FBS (Cytiva), and penicillin-streptomycin (Wako) at 37°C with 5% $CO_2$. To assess DT-cytotoxicity, 5,000 freshly trypsinized cells were seeded into each well of a 96-well microplate and incubated for 3 h. The medium was then replaced with DT-containing medium (DT BioAcademia, 01-517) or a mixture of DT and antibodies, followed by incubation for 48 h. The DT-antibody mixture was prepared by preincubating 500 ng/mL of antibody with 25 ng/mL of DT in room temperature for 1 h. Cell viability was measured using Cell Counting Kit 8 (Wako), and the $OD_{450}$ of each well was recorded using the GloMax Explorer Multimode Microplate Reader (Promega). The rate of cell death was calculated following previous publication (39).

## TT-GT1b receptor-binding inhibition assay

TT neutralization assay was done based on previously described protocol with slight modification (31, 32). TT0067 was used as a positive control (33). Twenty-five microliters per well of 20 µg/mL GT1b (Sigma Aldrich) diluted in methanol was coated on half-area 96-well plate and incubated for 1.5 h. After two times washing with 75 µL/well PBS-T, the plate was blocked using 50 µL/well Protein-Free Pierce Blocking Solution (Thermo) and then incubated for 1 h. Twenty micrograms per milliliter of His-Tag conjugated TT-binding domain were preincubated with same volume of 100 µg/mL TT-binding antibody for 1 h before being put on the well and incubated for 2 h. After three washes, HRP-conjugated anti-His Tag (BioLegend) was used as secondary antibody, and the detection was done in the same way as ELISA for antigen-binding assay.

## CHO-K1 cell-PT antibody neutralization assay

CHO-K1 cell line (Molecular Bacteriology Lab's stock) was cultured in Ham's F12K (Wako) supplemented with 10% FBS (Cytiva) and penicillin-streptomycin (Wako) at 37°C with 5% $CO_2$. Neutralization activity was assessed by using a modified visual clustering assay (40). Freshly trypsinized cells (10,000 cells/well) were seeded in a 96-well tissue culture plate and incubated overnight. The medium was then replaced with Ham's F12K supplemented with 1% FBS containing PT (Molecular Bacteriology Lab) or PT-antibody mixture. After 24 h of incubation, cell clustering morphology was evaluated by 5 independent observers using an Olympus U-TV2XC Microscope, assigning scores of 2 (mostly clustered), 1 (equivocal or partially clustered), or 0 (not clustered) per well. Duplicate wells were scored, and the mean and standard deviation (SD) were calculated. A score greater than 2 indicated a positive clustering response.

## Statistical analysis

Statistical analysis and data visualization were performed using Python, R, Gephi, or GraphPad Prism 9. Significance is determined by $P$-values <0.01.

## Source code

Methods to generate CDR annotations from AIRR-formatted files and to cluster the resulting pseudo sequences are available as part of the InterClone source code repository (https://gitlab.com/sysimm/interclone/).

## ACKNOWLEDGMENTS

We would like to thank Professor Eisuke Mekada, Professor Masato Okada, Professor Tetsuya Iida, Diego Diez, Shunsuke Teraguchi, Daisuke Motooka, and all members of the Genome Informatics and Molecular Bacteriology labs for valuable discussions in the development of this project. We also thank BIKEN Foundation (Taniguchi Scholarship), Fujii Scholarship, RIMD, and IFReC Core Experimental Facility for their support in conducting experiments. Graphical illustrations in Fig. 1 and the supplemental material were made using BioRender (https://biorender.com/).

This work was funded by Japan Agency for Medical Research and Development (AMED), Platform Project for Supporting Drug Discovery and Life Science Research (Basis for Supporting Innovative Drug Discovery and Life Science Research) under JP21am0101108.

## AUTHOR AFFILIATIONS

[1]Department of Genome Informatics, Research Institute for Microbial Diseases, Osaka University, Suita, Japan
[2]Graduate School of Medicine, Osaka University, Suita, Japan
[3]Department of Molecular Bacteriology, Research Institute for Microbial Diseases, Osaka University, Suita, Japan
[4]Immunology Frontier Research Center, Osaka University, Suita, Japan
[5]Center for Infectious Disease Education and Research, Osaka University, Suita, Japan
[6]Graduate School of Medical Safety Management, Jikei University of Health Care Sciences, Osaka, Japan

## AUTHOR ORCIDs

Dianita S. Saputri http://orcid.org/0000-0002-5538-0499
Dendi K. Nugraha http://orcid.org/0000-0002-6995-5695
Yasuhiko Horiguchi http://orcid.org/0000-0002-1592-5861

## FUNDING

| Funder | Grant(s) | Author(s) |
| --- | --- | --- |
| Japan Agency for Medical Research and Development (AMED) | JP21am0101108 | Daron M. Standley |
| BIKEN Foundation (BIKEN 財団) | Taniguchi Scholarship | Dianita S. Saputri |

## AUTHOR CONTRIBUTIONS

Dianita S. Saputri, Conceptualization, Data curation, Formal analysis, Investigation, Methodology, Validation, Visualization, Writing – original draft, Writing – review and editing | Hendra S. Ismanto, Conceptualization, Methodology, Writing – review and editing | Dendi K. Nugraha, Conceptualization, Methodology, Writing – review and editing | Zichang Xu, Resources, Software, Writing – review and editing | Yasuhiko Horiguchi, Supervision, Writing – review and editing | Shuhei Sakakibara, Supervision, Writing – review and editing | Daron M. Standley, Conceptualization, Data curation, Funding acquisition, Investigation, Methodology, Project administration, Supervision, Writing – review and editing

## DATA AVAILABILITY

Raw sequencing data and processed data sets generated in this study have been deposited in NCBI's Gene Expression Omnibus (41, 42) and are accessible through GEO Series accession number GSE238120.

## ETHICS APPROVAL

This study was approved by Osaka University Institutional Review Board. Informed consent was obtained from the participants after providing detailed information about the research objectives and potential implications.

## ADDITIONAL FILES

The following material is available online.

### Supplemental Material

**Fig. S1 to S3 and Tables S1 to S3 (mSystems00722-23-S0001.docx).** Additional experimental details and statistical analysis.
**Table S4 (mSystems00722-23-S0002.docx).** STORMS checklist.

### Open Peer Review

**PEER REVIEW HISTORY (review-history.pdf).** An accounting of the reviewer comments and feedback.

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
