## [Reviewer comments · mSystems]

Deciphering the antigen specificities of antibodies by clustering their complementarity determining region sequences

Dianita Saputri, Hendra Ismanto, Dendi Nugraha, Zichang Xu, Yasuhiko Horiguchi, Shuhei Sakakibara, and Daron Standley

Corresponding Author(s): Daron Standley, Osaka Daigaku

Review Timeline:

Submission Date:	August 7, 2023
Editorial Decision:	September 19, 2023
Revision Received:	October 5, 2023
Accepted:	October 6, 2023

Editor: Ileana Cristea

Reviewer(s): The reviewers have opted to remain anonymous.

Transaction Report:

DOI: <https://doi.org/10.1128/msystems.00722-23>

September 19, 2023

Prof. Daron M Standley
Osaka Daigaku
Genome Informatics
Yamadaoka 1-1
Suita, Osaka 5650871
Japan

Re: mSystems00722-23 (Deciphering the antigen specificities of antibodies by clustering their complementarity determining region sequences)

Dear Prof. Daron M Standley:

Thank you for submitting your manuscript to mSystems. We have completed our review and I am pleased to inform you that, in principle, we expect to accept it for publication in mSystems. However, acceptance will not be final until you have adequately addressed the reviewer comments.

Preparing Revision Guidelines

Please return the manuscript within 60 days; if you cannot complete the modification within this time period, please contact me. If you do not wish to modify the manuscript and prefer to submit it to another journal, please notify me of your decision immediately so that the manuscript may be formally withdrawn from consideration by mSystems.

Sincerely,

Ileana Cristea

Editor, mSystems

Journals Department
Reviewer comments:

Reviewer #1 (Comments for the Author):

This manuscript introduces a novel method to cluster antibodies sharing antigenic targets based on the CDR sequences. Authors applied this method to COVID-19 patient publicly available BCR data and Diphtheria-Tetanus-Pertussis (DTP)-vaccinated donors. Using this method, antibody expression and antigen binding assays demonstrated that the clusters exhibited 96% antigen purity compared to 82% purity achieved by assigning antigens to the same B cells using fluorescently labeled DTP antigen probes. This approach allows us to identify the specificities of many antibodies whose antigen targets are unknown, using a small fraction of antibodies with well-annotated binding specificities. The manuscript overall is well written, and studies are well controlled. Here are some comments:

1. As mentioned in the manuscript, the four DTP-vaccinated donors display quite a heterogeneity for antibody clusters as well as humoral immune responses. For example, donor 3's humoral immune response is weaker than that of other donors. It will be interesting to show whether the binding and neutralization abilities of antibodies derived from donor 3 differ from other donors whose immune responses are stronger after vaccination.
2. It will be helpful to indicate in Figure 5C which clusters these antibodies belong to. This will provide useful information for antibody binding, neutralization, and clustering information.
3. It's unclear whether the difference between the higher purity for COVID-19 patients data and DTP vaccination data is due to sample size difference or different basal level antibodies. As COVID-19 appeared as a new pathogen with no basal-level antibodies for the population, the majority of the population might have been vaccinated with DTP before this study, and the vaccination of DTP serves as a boost to their immune response. Please discuss this in the discussion section, as this will provide useful information.

Reviewer #2 (Comments for the Author):

This is a very interesting manuscript describing a novel approach of clustering B cell receptor repertoires based on their paratope (CDR), rather than on the epitope. The authors used two approaches to validate their approach, first using the publicly available database for SARS-CoV-2, and secondly applying their approach to immune sera from previously exposed human volunteers to DTaP or respective infections. The authors were able to confirm that in both cases their approach was able to allow them to cluster the repertoire based on the CDRs. Thus, the authors suggest, that when used in tandem with traditional approaches, this could help to characterize antibody repertoires with a higher resolution in a less complicated fashion.

Two quick comments that the authors may want to address:

1. There is no discussion of affinity maturation through somatic hypermutation in the manuscript, and one wonders if clustering the repertoire based on paratopes would also allow to predict clustering of high versus low affinity antibodies?
2. Secondly, as future vaccine target broadly neutralizing antibodies to increase the breadth of immune protection, one wonders how clustering based on the paratope might allow the selection for broadly neutralizing antibodies. It would be great to see the authors addressing these two points in their discussion.

Manuscript mSystems00722-23

Response to Reviewers

Dear Prof. Ileana Cristea

Thank you for giving us the opportunity to submit a revised version of the manuscript “*Deciphering the antigen specificities of antibodies by clustering their complementarity determining region sequences*” for publication in mSystems. We appreciate the time and effort that you and the reviewers dedicated in providing insightful comments and valuable feedback on our manuscript. Where feasible, we have incorporated the suggestions made by the reviewers and highlighted those changes within the manuscript. Below, please find our point-by-point responses to the reviewers’ comments and concerns. All page and line numbers refer to the revised manuscript file with tracked changes.

Reviewers’ Comments for the Author

Reviewer #1

The manuscript overall is well written, and studies are well controlled.

Author response: Thank you for reviewing our manuscript and giving constructive comments.

1. As mentioned in the manuscript, the four DTP-vaccinated donors display quite a heterogeneity for antibody clusters as well as humoral immune responses. For example, donor 3’s humoral immune response is weaker than that of other donors. It will be interesting to show whether the binding and neutralization abilities of antibodies derived from donor 3 differ from other donors whose immune responses are stronger after vaccination.

Author response: Thank you for pointing this out. Indeed, among the expressed antibodies from Clusters 1-25, Donor 3 only contributed to clusters that didn’t bind to DTP (Cluster 3, 7 and 20) and as a result, there were no neutralizing antibodies derived from this donor. We cannot say that there was an absolute absence of neutralizing antibodies in Donor 3 due to the sampling limitation of repertoire analysis and antibody expression, but these results suggest that Donor 3 was a weak responder. To highlight these observations, we have modified the main text at Results and Discussion part as follows:

Results (Line 241-243):

Indeed, among the expressed antibodies from Clusters 1-25, Donor 3 only contributed to clusters that didn’t bind to DTP antigen (Cluster 3, 7 and 20), indicating that Donor 3 is a weak responder to the vaccine.

Discussion (Line 300-302):

Moreover, we were able to identify a weak responder to the vaccine, highlighting the heterogeneity of human antibody responses to the same immune perturbation.

2. It will be helpful to indicate in Figure 5C which clusters these antibodies belong to. This will provide useful information for antibody binding, neutralization, and clustering information.

Author response: As suggested by the reviewer, we have modified Figure 5C by adding cluster information. Hopefully this can make it easier to see the antibody binding, neutralization, and clustering information in one place.

3. It's unclear whether the difference between the higher purity for COVID-19 patients' data and DTP vaccination data is due to sample size difference or different basal level antibodies. As COVID-19 appeared as a new pathogen with no basal-level antibodies for the population, the majority of the population might have been vaccinated with DTP before this study, and the vaccination of DTP serves as a boost to their immune response. Please discuss this in the discussion section, as this will provide useful information.

Author response: Although it is premature to generalize based on only two examples, we rather want to emphasize that 95 and 96% purities are very similar in spite of the differences in antigen, sample size, basal level of antibodies, and other factors. For this reason, we have not discussed the 1% difference in purity. We hope we have not misinterpreted the reviewer's comment.

Reviewer #2

This is a very interesting manuscript describing a novel approach of clustering B cell receptor repertoires based on their paratope (CDR)

Author response:

We appreciate the positive feedback very much.

1. There is no discussion of affinity maturation through somatic hypermutation in the manuscript, and one wonders if clustering the repertoire based on paratopes would also allow to predict clustering of high versus low affinity antibodies?

Author response:

Thank you for the suggestion to discuss more about the affinity maturation and somatic hypermutation. Here, we want to emphasize that the paradigm of paratope is more general than that of clonotype in terms of accommodating somatic hypermutation (SHM) in the CDRs. Unfortunately, predicting the affinities of antibodies is currently beyond the ability of our clustering software. However, we agree that this is a very important question to be addressed by additional analysis. To this end, we have added sentences to the Discussion section as follows:

Discussion (Line 290-293):

We also note that, by using paratope rather than clonotype as the representation of BCRs, we can include the effects of somatic hypermutation changes in the clusters. Therefore, features based on paratopes may allow prediction of antibody-antigen affinity.

2. Secondly, as future vaccines target broadly neutralizing antibodies to increase the breadth of immune protection, one wonders how clustering based on the paratope might allow the selection for broadly neutralizing antibodies. It would be great to see the authors addressing these two points in their discussion.

Author response: We agree that paratope clustering could potentially be used to identify antibody clusters that target antigens associated with different virus strains. It will be very interesting to examine whether such antibody clusters would serve as a biomarker for broad neutralization. We have added a sentence to the Discussion as follows:

Discussion (Line 321-325):

Paratope clustering may potentially be used to identify antibody clusters that target antigens associated with different viral or bacterial strains. In the future, it will be interesting to examine whether such antibody clusters can serve as biomarkers for broad neutralization upon vaccination.

October 6, 2023

Prof. Daron M Standley
Osaka Daigaku
Genome Informatics
Yamadaoka 1-1
Suita, Osaka 5650871
Japan

Re: mSystems00722-23R1 (Deciphering the antigen specificities of antibodies by clustering their complementarity determining region sequences)

Dear Prof. Daron M Standley:

Thank you for carefully addressing the reviewers' comments.

Congratulations! Your manuscript has been accepted, and I am forwarding it to the ASM Journals Department for publication. For your reference, ASM Journals' address is given below. Before it can be scheduled for publication, your manuscript will be checked by the mSystems production staff to make sure that all elements meet the technical requirements for publication. They will contact you if anything needs to be revised before copyediting and production can begin. Otherwise, you will be notified when your proofs are ready to be viewed.

If you would like to submit a potential Featured Image, please email a file and a short legend to msystems@asmusa.org. Please note that we can only consider images that (i) the authors created or own and (ii) have not been previously published. By submitting, you agree that the image can be used under the same terms as the published article. File requirements: square dimensions (4" x 4"), 300 dpi resolution, RGB colorspace, TIF file format.

We recognize that the video files can become quite large, and so to avoid quality loss ASM suggests sending the video file via <https://www.wetransfer.com/>. When you have a final version of the video and the still ready to share, please send it to mSystems staff at msystems@asmusa.org.

Sincerely,

Ileana Cristea
Editor, mSystems

Journals Department
E-mail: mSystems@asmusa.org